# Cognitive Behaviors that Enable Self-Improving Reasoners,
## *or,*
## Four Habits of Highly Effective STaRs

**Kanishk Gandhi**
Stanford University

**Ayush Chakravarthy**
Stanford University

**Anikait Singh**
Stanford University

**Nathan Lile**
SynthLabs

**Noah D. Goodman**
Stanford University

## Abstract

Test-time inference has emerged as a powerful paradigm for enabling language models to "think" longer and more carefully about complex challenges, much like skilled human experts. While reinforcement learning (RL) can drive self-improvement in language models on verifiable tasks, some models exhibit substantial gains while others quickly plateau. For instance, we find that Qwen-2.5-3B far exceeds Llama-3.2-3B under identical RL training for the game of Countdown. This discrepancy raises a critical question: what intrinsic properties enable effective self-improvement? We introduce a framework to investigate this question by analyzing four key *cognitive behaviors* — verification, backtracking, subgoal setting, and backward chaining — that both expert human problem solvers and successful language models employ. Our study reveals that Qwen naturally exhibits these reasoning behaviors, whereas Llama initially lacks them. In systematic experimentation with controlled behavioral datasets, we find that priming Llama with examples containing these reasoning behaviors enables substantial improvements during RL, matching or exceeding Qwen's performance. Importantly, the presence of reasoning behaviors, rather than correctness of answers, proves to be the critical factor — models primed with incorrect solutions containing proper reasoning patterns achieve comparable performance to those trained on correct solutions. Finally, leveraging continued pretraining with OpenWebMath data, filtered to amplify reasoning behaviors, enables the Llama model to match Qwen's self-improvement trajectory. Our findings establish a fundamental relationship between initial reasoning behaviors and the capacity for improvement, explaining why some language models effectively utilize additional computation while others plateau.[1]

> "The limits of my language mean the limits of my world."
> —Wittgenstein

## 1 Introduction

When humans encounter a difficult but solvable problem, we spend more time thinking deeply and deliberately to arrive at a solution. Remarkably, recent language models have begun demonstrating similar reasoning behaviors when trained to self-improve via reinforcement learning (Guo et al., 2025; Jaech et al., 2024). Training language models with reinforcement learning (RL) on verifiable problems is not a new approach (Zelikman et al., 2022; Havrilla et al., 2024; Hoffman et al., 2023), but older methods leveraging RL plateaued after a few iterations without exploring many effective ways to use test-time compute for thinking. In this work, we investigate the reasons behind this change in self-improvement capabilities, focusing on the presence of key cognitive behaviors in base language models.

We focus our investigation on two base models, Qwen-2.5-3B (Qwen et al., 2025) and Llama-3.2-3B (Grattafiori et al., 2024), which show striking differences when trained with

---

[1]Code available at https://github.com/kanishkg/cognitive-behaviors

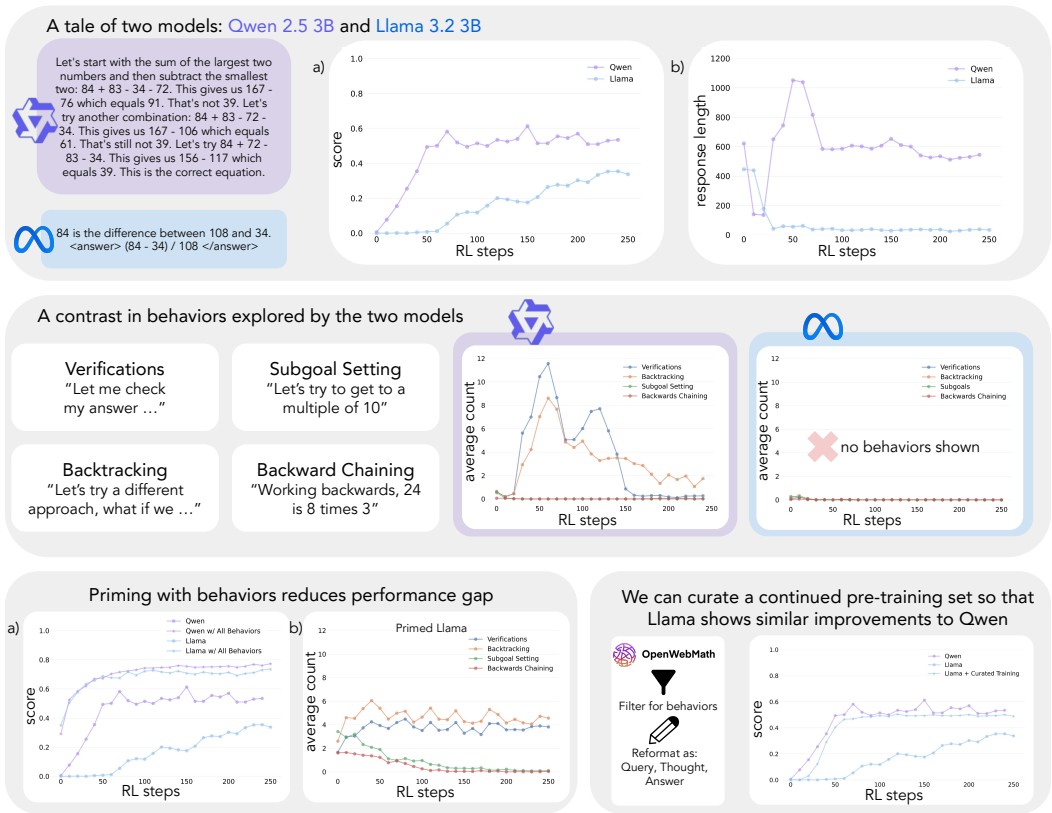

Figure 1: **Comparative analysis of Qwen-2.5-3B and Llama-3.2-3B models with RL on Countdown.** (Top) (a) Performance scores on Countdown for both models (b) Evolution of response lengths throughout RL training. (Middle) Emergence of specific reasoning characteristics as a function of training steps for Qwen-2.5-3B (left) and Llama-3.2-3B (right). (Bottom-Left) (a) Countdown performance when base models are primed with a synthetic dataset of desired reasoning behaviors; (b) Differential impact of RL on reasoning behaviors of the primed Llama3.2-3B: amplification of backtracking and verification contrasted with suppression of backward chaining and subgoal setting. (Bottom-Right) Comparative efficacy of teaching reasoning behaviors through fine-tuning on a curated OpenWebMath dataset, demonstrating that Llama's reasoning capabilities can be improved to match Qwen's through targeted training.

reinforcement learning on the game of Countdown (Countdown, 2024; Gandhi et al., 2024). While Qwen demonstrates substantial improvements in problem-solving ability, Llama[2] shows limited gains under an identical training process. What properties of the initial language model enable such improvements?

To systematically investigate this question, we develop a framework for analyzing *cognitive behaviors* that are useful for solving problems. We characterize four key cognitive behaviors: **verification** (systematic error-checking), **backtracking** (abandoning failing approaches), **subgoal setting** (decomposing problems into manageable steps), and **backward chaining** (reasoning from desired outcomes to initial inputs). These behaviors mirror how expert problem solvers approach difficult tasks — a mathematician verifies each step of a proof, backtracks when encountering contradictions, and breaks complex theorems into simpler lemmas. We examine these four behaviors because they can be used to represent search-based reasoning that goes beyond the typical "linear" reasoning shown by language models.

---

[2]For the remainder of this paper, we refer to Qwen2.5-3B as "Qwen" and Llama3.2-3B as "Llama", though full model names are also occasionally used. All other models are specified with complete designations.

While many other cognitive behaviors exist, we begin with these as they are well-defined and easily identifiable in model outputs.

Our initial analysis reveals that Qwen naturally exhibits these reasoning behaviors, particularly verification and backtracking, while Llama lacks them. This observation motivates our central hypothesis: certain reasoning behaviors in the initial policy are necessary for efficiently utilizing increased test-time compute through extended reasoning sequences. We test this hypothesis through interventions on the initial model. First, we demonstrate that priming Llama with synthetic reasoning traces containing these behaviors, especially backtracking, enables substantial improvements during RL, matching Qwen's performance trajectory. Second, these gains persist even when primed on incorrect solutions, if they exhibit proper reasoning patterns, suggesting that the presence of reasoning behaviors, rather than access to correct solutions, is the critical factor enabling successful self-improvement. Third, by curating pretraining data from OpenWebMath (Paster et al., 2023) that emphasizes these reasoning behaviors, we show that targeted modification of the pretraining distribution can successfully induce the behavioral patterns necessary for efficient use of test-time compute — the improvement trajectory of Llama matches that of Qwen.

Our investigation reveals a strong relationship between a model's initial reasoning behaviors and its capacity for improvement. This connection helps explain why some language models discover effective ways to use additional compute while others plateau. Understanding these dynamics may be key to developing AI systems that can meaningfully improve their problem-solving abilities.

## 2 Related Work

Recent approaches to improving reasoning capabilities in language models can be broadly categorized into three complementary directions: external search methods that leverage multiple samples, in-context search that enables models to reason over their own outputs, and reinforcement learning approaches that allow models to discover reasoning strategies autonomously.

**External Search for Reasoning.** Recent work has shown that language models can significantly improve their performance on complex tasks when given additional inference-time compute. Snell et al. (2024) systematically explore this space by developing various methods to search through reasoning trajectories. These approaches range from simple parallel sampling (Li et al., 2024; Brown et al., 2024) to more sophisticated methods using verifiers or process reward models (PRMs) (Lightman et al., 2023; Yao et al., 2023). Some researchers have taken this further by using the search process itself to improve the underlying reasoning model (Wang et al., 2023; Luo et al., 2024). However, these methods typically operate without awareness of previously explored solutions, limiting their efficiency through redundant exploration.

**In-Context Search and Self-Improvement.** In contrast to external search methods, another line of research focuses on enabling models to search sequentially in language. This has been achieved through various approaches such as 1) in-context examples (Gandhi et al., 2023), 2) finetuning on linearized search traces (Gandhi et al., 2024; Lehnert et al., 2024), and 3) training on self-correction examples (Ye et al., 2024; Qu et al., 2025; Kumar et al., 2024; Hwang et al., 2024). Recent work by Schultz et al. (2024) bridges the gap between in-context and external search methods, demonstrating improved performance on strategic games (cf. Xiang et al., 2025). While effective, these approaches often require careful engineering of training data to incorporate desired behaviors like self-correction and backtracking.

**Reinforcement Learning for Reasoning.** The prospect of models autonomously discovering effective reasoning strategies has motivated significant research in reinforcement learning approaches. Early work in teaching language models to reason with verifiable outcomes explored various RL methodologies, from off-policy and batch methods (Zelikman et al., 2022; Havrilla et al., 2024; Hoffman et al., 2023) to on-policy approaches (Zelikman et al., 2024; Kazemnejad et al., 2024; Cui et al., 2025). These methods differ in their approaches to credit assignment in reasoning trajectories. A notable breakthrough

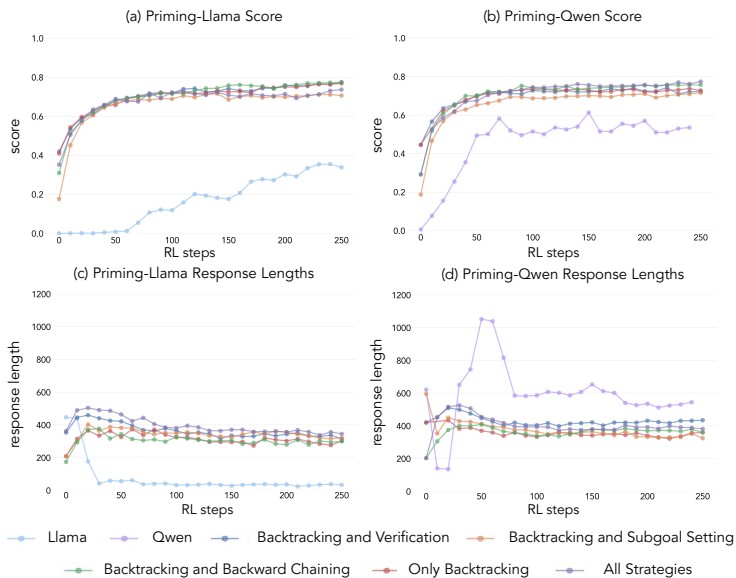

Figure 2: **The effects of priming with different cognitive behaviors.** (a, b) Performance comparison on the Countdown task between Llama-3.2-3B, Qwen-2.5-3B and their primed variants, illustrating the influence of reasoning behavior priming on scores. (c, d) Response length analysis for standard and primed Llama and Qwen models, showing how priming influences reasoning length.

came with Deepseek-R1 (Guo et al., 2025; Jaech et al., 2024), which demonstrated that even a simplified version of PPO (Schulman et al., 2017) called GRPO (Shao et al., 2024) could lead to significant improvements and the emergence of in-context search behaviors. Recent analyses (Li et al., 2025), Wu et al. (2024), Liu et al. (2025), Ye et al. (2025) and Yeo et al. (2025) have begun to unpack these results, revealing that supervised fine-tuning with long, structured chains of thought enhances both the efficiency and performance of RL compared to shorter reasoning chains. However, a crucial question remains unanswered: why do some models successfully learn through RL while others fail to improve? (Yeo et al., 2025)briefly discuss this issue by analyzing the frequency of phrases related to reflective reasoning in the pretraining dataset but their analysis remains limited. Our work addresses this gap by investigating the essential properties of initial models that enable successful reinforcement learning of reasoning behaviors.

## 3 Identifying and Engineering Self-Improving Behavior

### 3.1 Initial Investigation: A tale of two models

We begin by investigating a surprising observation: language models of comparable size but from different families show markedly different capacities for improvement through reinforcement learning. The Countdown game serves as our primary testbed — a mathematical puzzle where players must combine a set of input numbers using the four basic arithmetic operations $(+, -, \times, \div)$ to reach a target number. For example, given the numbers 25, 30, 3, 4 and a target of 32, the solution involves combining these numbers through a series of operations to reach exactly 32: $(30 - 25 + 3) \times 4$.

We selected Countdown for our analysis because it demands mathematical reasoning, planning, and search strategies that mirror key aspects of general problem-solving similar to other reasoning domains such as math. Unlike more complex domains, Countdown offers a restricted search space that enables tractable analysis while still requiring sophisticated reasoning. Additionally, success in the game depends more on problem-solving abilities than mathematical knowledge, compared to other mathematical tasks where domain knowledge might confound the evaluation of reasoning capabilities.

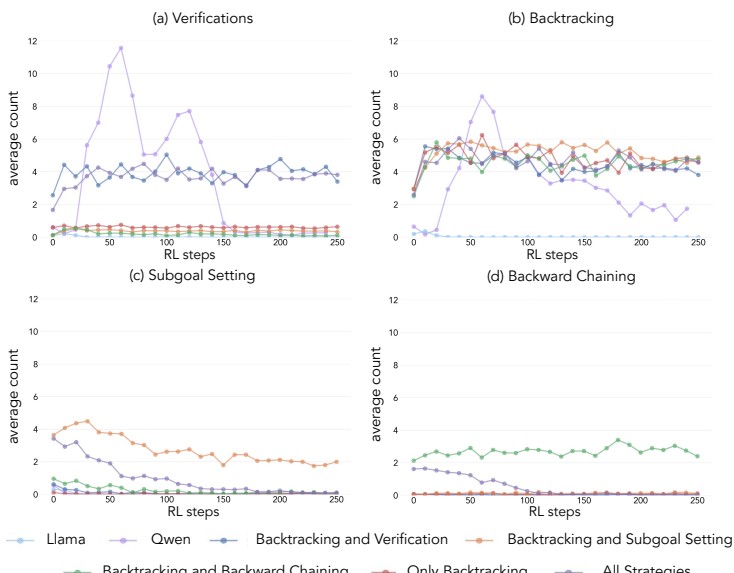

Figure 3: **Analysis of four key reasoning behaviors with Llama-3.2-3B, Qwen-2.5-3B, and primed versions of Llama-3.2-3B.** Plots show mean frequency of (a) solution verification steps, (b) problem-solving backtracking instances, (c) explicit subgoal setting, and (d) backward chaining reasoning approaches across different tasks.

We use two base models to contrast how learning varies across model families: Qwen-2.5-3B and Llama-3.2-3B. Our reinforcement learning experiments build on the VERL library (Sheng et al., 2024), utilizing the TinyZero (Pan et al., 2025) implementation. We train models with PPO (Schulman et al., 2017) for 250 steps, sampling 4 trajectories per prompt. We chose PPO over alternatives like GRPO (Shao et al., 2024) and REINFORCE (Williams, 1992; Hu, 2025; Ahmadian et al., 2024) as it demonstrated superior stability across hyperparameter settings, though performance was anecdotally similar across algorithms.

The results reveal strikingly different learning trajectories. Though both models start at a similar, low performance on the task, Qwen demonstrates a qualitative shift around step 30, characterized by significantly longer responses and improved accuracy (Fig. 1 (top)). By the end of training, Qwen achieves approximately 60% accuracy, substantially outperforming Llama's 30%. Later in training, we observe an interesting change in Qwen's behavior: the model transitions from explicit verification statements in language, "8*35 is 280 which is too high" to implicit solution checking, where the model sequentially tries different solutions until it finds the right answer without using words to evaluate its own work.

This contrast raises a fundamental question: what underlying capabilities enable successful reasoning-based improvement? To answer this, we need a systematic framework for analyzing cognitive behaviors.

## 3.2 A Framework for Analyzing Cognitive Behaviors

To understand these divergent learning trajectories, we develop a framework for identifying and analyzing key behaviors in model outputs. We focus on four fundamental behaviors: (1) Backtracking or the explicit revision of approaches when errors are detected (e.g., "This approach won't work because..."), (2) Verification or the systematic checking of intermediate results (e.g., "Let's verify this result by..."), (3) Subgoal Setting, where a complex problem is broken down into manageable steps (e.g., "To solve this, we first need to..."), and (4) Backward Chaining, where in a goal-directed reasoning problem, the solution works backwards from a desired outcomes (e.g., "To reach the target of 75, we need a number divisible by...").

We selected these behaviors because they represent problem-solving strategies that deviate from the linear, monotonic reasoning patterns commonly observed in language models. These behaviors enable more dynamic, search-like reasoning trajectories where solutions

can evolve non-linearly. While this set is not exhaustive, these behaviors were chosen because they are easy to identify and they align naturally with human problem-solving strategies in both the Countdown game and broader mathematical reasoning tasks like proof construction.

Each behavior can be identified by its pattern in the reasoning tokens. Backtracking is seen as token sequences that explicitly contradict and replace previous steps, verification produces tokens that compare outcomes against solution criteria, backward chaining generates tokens that construct solution paths from the goal to the initial state, and subgoal setting explicitly proposes an intermediate step to target on the path to the final goal. We develop a classification pipeline using GPT-4o-mini[3] to reliably identify these patterns in the outputs of the model (see App. F for analysis of agreement with humans and larger models).

### 3.3 The Role of Initial Behaviors in Self-Improvement

Applying this framework to our initial experiment reveals a key insight: Qwen's dramatic performance improvements coincided with the emergence of cognitive behaviors, particularly verification and backtracking (Fig. 1 (middle)). Llama, in contrast, showed minimal evidence of these behaviors throughout training.

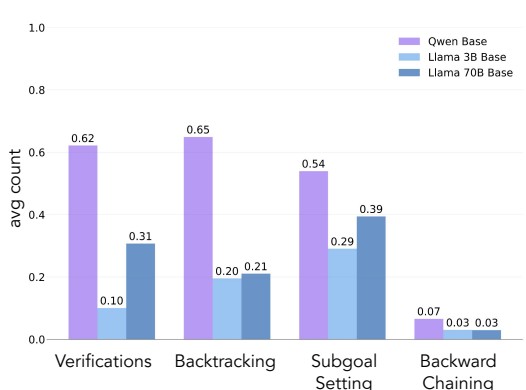

To better understand this disparity, we analyzed the baseline reasoning patterns across three models: Qwen-2.5-3B, Llama-3.2-3B, and Llama-3.1-70B. Our analysis revealed that Qwen-2.5-3B naturally exhibits substantially higher rates of all four behaviors compared to both Llama variants (Fig. 4). While the larger Llama-3.1-70B showed generally increased activation of these behaviors compared to Llama-3.2-3B, this improvement was notably uneven — back-

Figure 4: **Exploration of different reasoning behaviors in base models**. An analysis with Qwen2.5-3B, Llama3.2-3B, and Llama3.1-70B on Countdown.

tracking, in particular, remained limited even in the larger model. These observations suggest two insights: 1) certain cognitive behaviors in the initial policy may be necessary for models to effectively utilize increased test-time compute through extended reasoning sequences 2) increased model scale can improve the contextual activation of these behaviors. This pattern is particularly significant because reinforcement learning can only amplify behaviors that appear in successful trajectories — making these initial behavioral capabilities a precondition for effective learning.

### 3.4 Intervening on initial behaviors

Having established the importance of cognitive behaviors in base models, we next investigate whether we could artificially induce these behaviors through targeted interventions. Our hypothesis is that by creating variants of base models that selectively exhibit specific cognitive behaviors before RL training, we can better understand which behavioral patterns are crucial for enabling effective learning.

We begin by curating seven distinct priming datasets using Countdown problems. Five of these datasets emphasize different behavioral combinations: all strategies combined, backtracking only, backtracking with verification, backtracking with subgoal setting, and backtracking with backward chaining. We generate these datasets using Claude-3.5-Sonnet[4], leveraging its ability to produce reasoning trajectories with precisely specified behavioral characteristics. While Claude does not always produce the correct answer (see Fig. 9), it consistently demonstrates the requested reasoning patterns, providing clean behavioral

---

[3]We chose GPT-4o-mini to balance inference cost and model capability

[4]We chose Claude-3.5-Sonnet as it reliably followed instructions to show the desired behaviors we wanted in the reasoning trajectories.

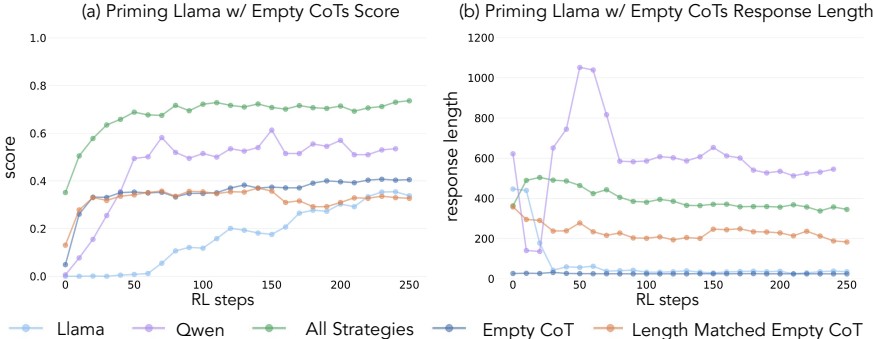

Figure 5: **Impact of Empty Chain-of-Thought Priming on Model Performance and Response Characteristics** Comparative analysis of (a) performance scores on the Countdown task and (b) response length distributions across different model configurations: baseline models (Llama3.2-3B, Qwen2.5-3B), unprimed conditions, length-matched empty CoT priming, and all-strategies-primed Llama3.2-3B.

primitives for our analysis. To verify that improvement stems from specific cognitive behaviors rather than simply increased computation time, we introduce two control conditions: 1) an empty chain-of-thought, where the thoughts are empty, "*<think></think>*" and 2) a chain-of-thought where the thoughts have the same length as those of the all-strategies control, but filled with placeholder tokens, "*<think>. . . . . </think>*". We also create a variant of our all-strategies dataset containing only incorrect solutions while maintaining the desired reasoning patterns. This variant allows us to disentangle the importance of cognitive behaviors from the accuracy of solutions.

**Priming with different behaviors**   When initialized with datasets containing backtracking behaviors, both Llama and Qwen demonstrate substantial improvements through RL training (Fig. 2). Behavioral analysis reveals that RL selectively amplifies empirically useful behaviors while suppressing others (Fig. 3). For instance, in the all-strategies condition (Fig. 1 (bottom-left)), the models retain and strengthen backtracking and verification while diminishing backward chaining and subgoal setting. However, the suppressed behaviors (backward chaining and subgoal setting) when paired only with backtracking persist through training.

**Testing Behavioral Necessity.**   When primed with the empty chain-of-thought controls, in both conditions, the models performance is comparable to the base Llama model (≈30-35%; see Fig. 5), demonstrating that the mere allocation of additional tokens without the inclusion of cognitive behaviors fails to enable effective use of test-time compute. Further, training with an empty chain-of-thought has a detrimental effect, where the Qwen model stops exploring the behaviors. This suggests that these cognitive behaviors are specifically necessary for models to make productive use of extended computation through longer reasoning sequences.

**Behaviors versus Correctness.**   Surprisingly, models primed with incorrect solutions, but with the right behaviors achieve identical performance to those trained on datasets with correct solutions (Fig. 6). This suggests that the presence of cognitive behaviors in the priming data, rather than the access to correct solutions, is the crucial factor enabling successful self-improvement through reinforcement learning. This extends prior work demonstrating learning from corrupted reasoning trajectories (Li et al., 2025) in a significant way — we show that reasoning patterns from weaker models can effectively bootstrap the learning process to build more capable ones, suggesting that the presence of cognitive behaviors matter more than just the correctness of the outcomes.

The above results show that certain cognitive behaviors are necessary for self-improvement. However, our priming method for inducing behaviors in the initial model was domain-specific, relying on the Countdown game. This may adversely impact the generalization of

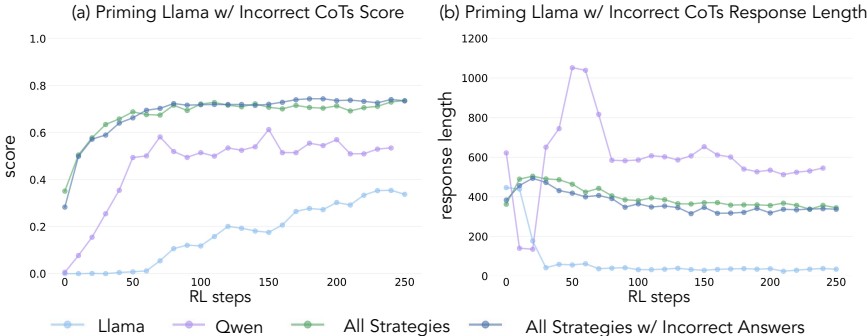

Figure 6: **Effects of Incorrect Chain-of-Thought Priming on Model Performance and Output Characteristics** Evaluation of (a) Countdown task performance scores and (b) response length distributions comparing four conditions: Llama3.2-3B and Qwen2.5-3B base models, Llama3.2-3B with correct all-strategy priming, and Llama3.2-3B primed with incorrect reasoning examples.

the resulting reasoning. Can we instead enable self-improvement by modifying a model's pretraining distribution to increase the frequency of beneficial reasoning behaviors?

**Behavioral Frequencies in Pretraining Data.** We first analyze the natural frequency of cognitive behaviors in pre-training data, focusing on OpenWebMath (Paster et al., 2023), and FineMath (Allal et al., 2025), which are constructed specifically for mathematical reasoning. Using Qwen-2.5-32B as a classifier[5], we analyze 200,000 randomly sampled documents for the presence of our target behaviors. Even in this math-focused corpus, cognitive behaviors such as backtracking and verifications appear infrequently, suggesting that standard pretraining provides limited exposure to these crucial patterns (see Fig. 7).

### 3.5 Selectively amplifying behaviors in pretraining data

**Behavioral Augmentation of the Pretraining Data.** To test whether artificially increasing exposure to cognitive behaviors enhances the potential for self-improvement, we develop a targeted continued pretraining dataset from OpenWebMath. First, using Qwen-2.5-32B as a classifier, we analyze mathematical documents from the pretraining corpus for the presence of our target reasoning behaviors. This allows us to create two contrasting sets: one with the cognitive behaviors and a control set that shows minimal evidence of these behaviors. We then use Qwen-2.5-32B to rewrite each document in the set into a structured question-thought-answer format, preserving the natural presence or absence of cognitive behaviors from the source documents. The final pretraining datasets each contain a total of 8.3 million tokens. This approach allows us to isolate the impact of reasoning

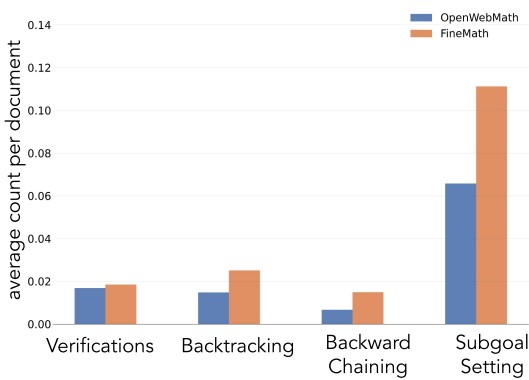

Figure 7: **Behaviors present in Math Pretraining Datasets**. An analysis of the behaviors present in 200,000 randomly sampled documents of OpenWebMath and FineMath. We measure the average count of the behaviors in each document.

behaviors while controlling for the format and amount of mathematical content during pre-training.

---

[5]We use Qwen-2.5-32B to balance throughput, capability and cost.

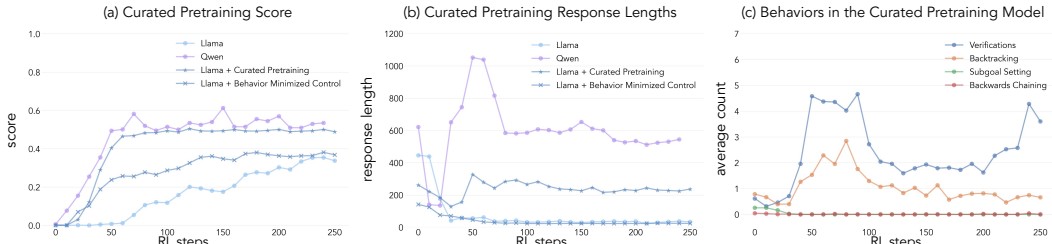

Figure 8: **Impact of curated pretraining on model performance and behavior.** (a) Comparison of performance scores across base models (Llama-3.2-3B, Qwen-2.5-3B) and Llama variants with curated pretraining versus behavior-minimized control. (b) Evolution of response lengths during training for each model configuration. (c) Emergence and development of specific cognitive behaviors in the curated pretraining model over training steps.

After pre-training Llama-3.2-3B on these datasets and applying reinforcement learning, we observe that: 1) the behavior-enriched model achieves performance comparable to Qwen, while the control model shows limited improvement (Fig. 8a) and 2) behavioral analysis of the trained models reveals that the behavior-enriched variant maintains high activation of reasoning behaviors throughout training, while the control shows behaviors similar to the base Llama model (Fig. 8c). These results demonstrate that targeted modification of pre-training data can successfully induce the cognitive behaviors necessary for effective self-improvement through reinforcement learning.

## 4 Discussion

We have found that a model's initial exploration of cognitive behaviors – particularly its tendency toward verification, backtracking, subgoal setting, and backward chaining – plays a crucial role in enabling self-improvement. Models that naturally exhibit these reasoning behaviors (such as Qwen-2.5-3B) show dramatically better improvement through RL compared to models lacking these behaviors (such as Llama-3.2-3B). Priming models with cognitive behaviors, by a small amount of finetuning, enabled significant performance gains even in models that initially lack these capabilities. Remarkably, this holds even when primed with incorrect solutions that exhibit the target behavioral patterns, suggesting that cognitive behaviors matter more than solution accuracy. Together these results indicate that the presence of cognitive behaviors is a *causal* factor enabling self-improvement through RL. Our initial priming experiments used data based on the Countdown game for training, possibly limiting generality. We thus developed a more diverse behavior-enriched training dataset derived from OpenWebMath. Training Llama on this set then led to self-improvement comparable to Qwen, demonstrating that the capacity for improvement can be engineered through careful curation of pretraining data.

When humans try to solve problems that are difficult but not unsolvable for them, they exhibit certain behaviors that support the problem-solving processes, structuring search over the space of possible solutions to a problem. These *cognitive behaviors* are usually sequential, deliberate and dependent on the problem space (Simon & Newell, 1971). Correspondingly, the cognitive behaviors that are amplified or suppressed during RL training are likely to be highly dependent on the tasks and environments being optimized for. In our studies using Countdown, backtracking and verification were the most critical. This raises important questions about the patterns that enable self-improvement in tasks such as coding, game play or creative writing. We believe that the principle described here will extend to other domains, but future work should explore how task-specific constraints interact with cognitive behaviors. Further, the cognitive behaviors specified in this work are not exhaustive; other behaviors are worth exploring, such as making analogies (Mitchell, 2021) and identifying one's existing state of knowledge (Metcalfe, 1986).

In conclusion, our findings show how cognitive behaviors enable language models to show self-improvement — effectively using increased test-time compute to solve increasingly more challenging problems. Humanity has a rich inheritance of cognitive behaviors that

enable effective reasoning. Future artificial intelligence may go beyond learning to use these existing behaviors – it may discover new ones, potentially revealing entirely new approaches to reasoning and computation.

## Acknowledgments

We would like to thank Charlie Snell, Eric Zelikman, Archit Sharma, Rafael Rafailov, Alex Havrilla, Ulyana Piterburg, Chase Blagden, Dimitris Papailiopoulos, and Jan-Philipp Fränken for their discussions and support. KG was supported by an HAI-SAP Grant and an NSF Expeditions grant; AS was supported by the NSF GRFP Fellowship. Compute for this work was provided by Synthlabs, Stanford Marlowe and the Toyota Research Institute (TRI).

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

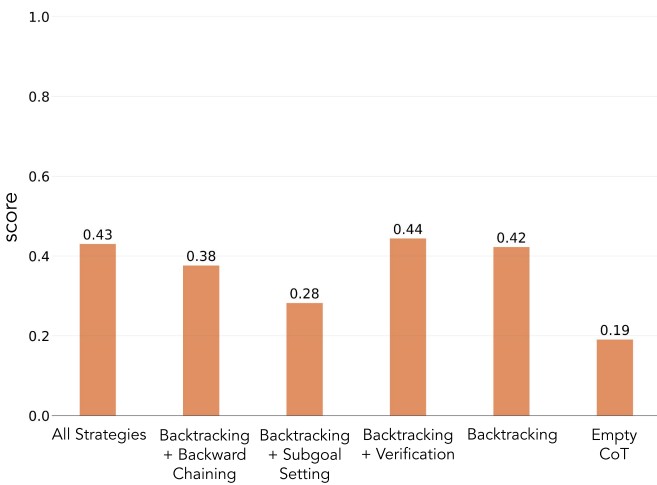

Figure 9: **Scores of the priming datasets generated with Claude.** Analyzing the average scores of Claude when instructed to solve Countdown employing different cognitive behaviors.

## A    Data Generation

For our experimental setup, we implemented the Countdown task according to the methodology outlined by Gandhi et al. (2024) and Pan et al. (2025). The task consists of an equal distribution (50-50 split) between 3-digit and 4-digit Countdown problems. Both the starting numbers and the target number were generated using random sampling to ensure variability across trials while maintaining consistent difficulty parameters. This randomization approach allowed us to assess mathematical problem-solving capabilities across a representative range of Countdown games.

## B    Priming

To prepare our supervised fine-tuning (SFT) data, we use Claude 3.5 Sonnet (claude-3-5-sonnet-20241022) to generate reasoning trajectories. We developed five distinct SFT datasets, each designed to capture different cognitive behaviors:

1. Backtracking Only: This dataset focuses exclusively on the backtracking strategy, where the model explores solution paths and retreats when encountering dead ends.

2. Backtracking with Answer Verification: In addition to backtracking, this dataset incorporates answer verification, where the model checks its intermediate solutions with the target number.

3. Backtracking with Subgoal Setting: This dataset combines backtracking with explicit subgoal setting, where the model breaks down complex problems into manageable intermediate steps.

4. Backtracking with Backward Chaining: This dataset demonstrates backward chaining with backtracking, where the model works backward from the goal state to the initial state.

5. All Strategies: This comprehensive dataset incorporates all four reasoning strategies mentioned above.

To control specific reasoning behaviors across each dataset, we implemented customized system prompts that explicitly guide Claude toward the desired reasoning patterns. These

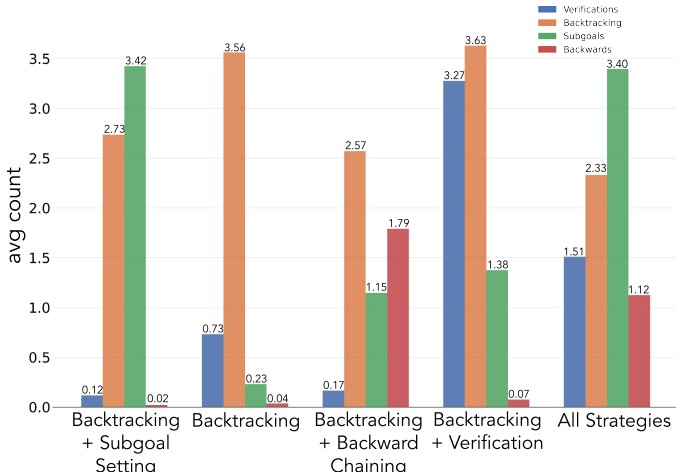

Figure 10: **Behaviors present in priming datasets.** Analyzing the average counts of behaviors when Claude is instructed to only show specific behaviors in its thoughts.

system prompts [6] were carefully crafted to elicit distinct problem-solving approaches while maintaining consistency across datasets (see Fig. 10 for an analysis of behaviors present). To create datasets rich in only the targeted behaviors and determine causal efficacy, we instructed Claude to use exclusively the specified cognitive behavior while prohibiting all others. For example, in the "Backtracking Only" condition, we emphasized through the system prompt that Claude could only use backtracking and was explicitly not permitted to verify answers, set subgoals, or work backwards from the target. For each of the five datasets, we selected 1,200 unique Countdown games as our problem set. We prompted Claude to solve these games while adhering to the specified reasoning strategy for each dataset. We then evaluated the accuracy of Claude's solution trajectories, with results presented in Fig. 9.

We provide the SFT hyperparameters in Tab. 1.

| Data & Model | |
| --- | --- |
| Training/Validation Dataset Size | 1000 / 200 |
| Context Window | 2048 |
| Training/Validation Batch Size | 64 / 64 |
| Optimization | |
| Optimizer | AdamW (Loshchilov & Hutter, 2019) |
| Peak Learning Rate | 1e-5 |
| Warmup | Linear (5% of total steps) |
| Annealing | Cosine |
| Total Epochs | 5 |

Table 1: SFT Hyperparameters

## C   Reinforcement Learning

We provide the PPO training hyperparameters in Tab. 2.

---

[6]All prompts are available in our Github repository: https://github.com/kanishkg/cognitive-behaviors.

| Data & Model | |
| --- | --- |
| Training/Validation Batch Size | 256 / 1312 |
| Context Window | 256 prompt + 1024 response tokens |
| **Optimization** | |
| Actor Learning Rate | 1e-6 |
| Critic Learning Rate | 1e-5 |
| KL Coefficient | 0.001 |
| PPO Mini-batch Size | 128 |
| Number of Rollouts | 4 |
| Rollout Temperature | 1.0 |
| **Reward Structure** | |
| Correct Answer | 1.0 |
| Incorrect Answer | 0.0 |
| Correct Format Bonus | 0.1 |

Table 2: PPO Training Hyperparameters

## D   Metrics

Next, we describe our behavioral evaluation metrics. The 4 behavioral metrics we track are:

1. Average Backtracking Count
2. Average Verification Count
3. Average Backward-Chaining Count
4. Average Subgoal-Setting Count

We generate samples from the trained model at using a temperature of 1.0, and a maximum of 1024 tokens. Each of these metrics track the average occurrence of each reasoning behavior. We develop a classification pipeline using GPT 4o-mini (the classifier model).

Our classification pipeline asks the classifier model 4 questions per reasoning trajectory. In each question, we provide the classifier model examples of each behaviors, so for instance for answer verification in Countdown, we add examples like "This sequence results in 1, which is not equal to 22" in the prompt.

For each reasoning behavior, we ask the classifier model to count up and report the number of distinct occurrences. We sample 512 tokens from the classifier model with temperature set to 0 for reproducibility.

## E   Pretraining Data Interventions.

**Analyzing Frequency in Pretrained Data.**   To investigate the natural frequency distribution of various reasoning behaviors in the OpenWebMath (Paster et al., 2023) and FineMath (Allal et al., 2025) datasets, we used the Qwen2.5-32B model as the classifier, deployed via vLLM (Kwon et al., 2023) for efficient inference. The analysis encompassed 200,000 randomly sampled documents from both datasets. For each document, the classifier model was prompted with examples demonstrating specific cognitive behaviors, following the classification framework described in App. D. For instance, to identify instances of Subgoal Setting, we provided examples such as: "To solve this system of equations, let's first isolate x in the first equation, then substitute it into the second." To ensure reproducibility, all classification runs were conducted with a temperature setting of 0. Each document was processed with a generation length of 1024 tokens per cognitive behavior. This methodical approach enabled us to quantitatively assess the prevalence of different cognitive behaviors across the mathematical content in both datasets.

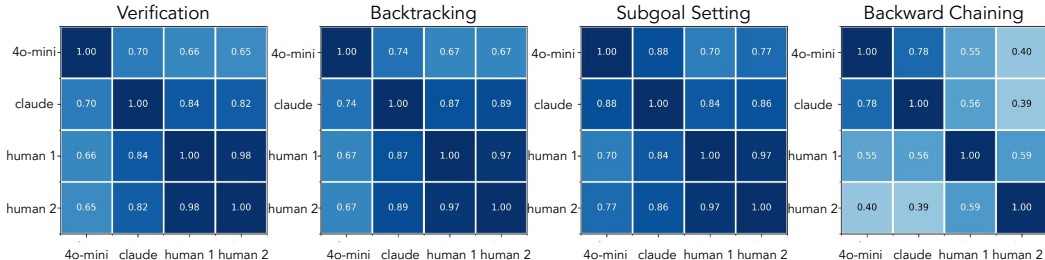

Figure 11: **Inter-rater reliability for counting and classifying behaviors.** Intraclass Correlation Coefficient (ICC3) values for (a) Verification (b) Backtracking (c) Subgoal Setting (d) Backward chaining for labels from GPT-4o-mini, Claude, and two human raters.

**Curating Pretraining Data.**    We write similar scripts using the same classification pipeline to perform two data preprocessing steps:

1. **Isolating Reasoning Behaviors.** We leverage our classification pipeline to systematically identify and extract passages exhibiting specific cognitive behaviors from the source datasets. For each document, the classifier determines whether each of the cognitive behaviors is exhibited in that document. These binary evaluations, are then used to populate two datasets - the behavior curated training, and behavior-minimized control training datasets.

2. **Re-formatting for Training.** We process the data into a structured format utilizing XML tags to delineate question, thinking process, and answer components while maintaining the integrity of the original question content. During the paraphrasing process, we implement first-person language specifically within the thinking sections to accurately represent cognitive processes. For the behavior-curated training dataset, we incorporate explicit instructions to ensure the paraphrased thoughts contain the targeted cognitive behaviors under investigation. Conversely, for the behavior-minimized dataset, we omit these specific instructions to establish an appropriate control condition.

## F    Interrater Reliability for Classifying Behaviors

To assess the reliability of classification and counting of behaviors from GPT-4o-mini, we conducted an inter-rater reliability analysis using the Intraclass Correlation Coefficient (ICC3). We compared agreement of GPT-4o-mini, Claude, and two human annotators — of four key problem-solving behaviors: verification, backtracking, subgoal setting, and backward chaining. We randomly sample 100 reasoning trajectories from different conditions and trained models to measure agreement.

GPT-4o-mini showed (see Fig. 11) strong agreement with other raters across most behaviors, though with varying levels of consistency. For verification, GPT-4o-mini achieved substantial agreement with both Claude (ICC3 = 0.70) and human raters (ICC3 = 0.66 and 0.65). For backtracking, GPT-4o-mini maintained robust reliability with Claude (ICC3 = 0.74) and slightly lower but consistent agreement with human raters (ICC3 = 0.67). Subgoal setting showed the strongest consistency, with GPT-4o-mini achieving high agreement with Claude (ICC3 = 0.88) and human raters (ICC3 = 0.70 and 0.77). Backward chaining demonstrated lower agreement levels, with GPT-4o-mini showing stronger correlation with Claude (ICC3 = 0.78) than with human raters (ICC3 = 0.55 and 0.40). These results suggest that GPT-4o-mini can reliably identify and classify most cognitive behaviors.

## G    Test-time Scaling.

We test how model performance scales with different token budget constraints at inference time. By varying the maximum allowed token length from $2^7$ (128) to $2^{10}$ (1024) tokens, we

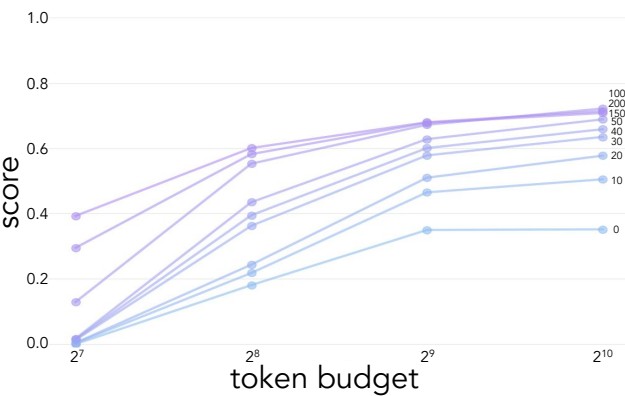

Figure 12: **Test-time Scaling by Varying Token Budgets during Inference.** The graph shows how different training checkpoints of the Llama model, fine-tuned with RL using the all-strategies primed model when tested with different maximum token lengths (128, 256, 512, and 1024 tokens). This analysis demonstrates how model performance scales with available computational resources at inference time.

analyzed the relationship between computational resources and model performance. We test training checkpoints ranging from 0 to 200 steps of PPO training for the all-strategies primed Llama model. Performance consistently improves with larger token budgets across all checkpoints, but with diminishing returns after 512 tokens (see Fig. 12). Later checkpoints (150-200 steps) show stronger performance even with restricted token budgets, achieving scores of ≈0.4 with just 128 tokens compared to near-zero performance for early checkpoints under the same constraints. This suggests that training helps the model learn to use limited token budgets more efficiently — more train-time interactions lead to more efficient test-time performance (Jones, 2021). The scaling behavior also reveals an interesting convergence pattern - while early checkpoints (0-20 steps) show nearly log-linear scaling with token budget increases, later checkpoints exhibit more logarithmic scaling, suggesting they have learned to make better use of additional tokens. By 1024 tokens, the best checkpoints achieve scores around 0.7, with relatively small gaps between checkpoints after 100 training steps.

## H    Transfer of Behaviors to Other Domains.

To evaluate the generalization of learned cognitive behaviors, we investigated whether models trained to exhibit specific problem-solving strategies in mathematical reasoning could transfer these behaviors to other data distributions. First, we compared two variants: a model trained with a curated pretraining dataset designed to amplify cognitive behaviors in mathematical reasoning, and a control model trained on behavior-minimized data on 200 questions from GPQA (Google Proof Question Answering; Rein et al. (2024)). The results (see Fig. 13) demonstrate significant transfer of learned behaviors, with the curated pretraining model exhibiting substantially higher frequencies of all four cognitive strategies (verification, backtracking, subgoal setting, and backward chaining) when solving general knowledge questions. Most notably, subgoal setting showed the strongest transfer effect, with an average of 6.5 instances per question in the curated model compared to 0.7 in the control. This suggests that cognitive behaviors amplified in mathematical reasoning can generalize effectively to broader question-answering contexts. It should be noted that there is no significant performance difference between the two models (both score about ≈ 12%). This performance parity despite behavioral differences suggests is because the model is limited in its forward inference capabilities rather than its problem-solving strategy — in other words, the model can learn to approach problems more systematically, but still struggles with the basic reasoning and knowledge needed to arrive at correct answers.

**RL training on Countdown transfers to other domains.**    Next, we tested a Qwen model trained with RL on Countdown and the base Qwen model on 200 questions from GPQA

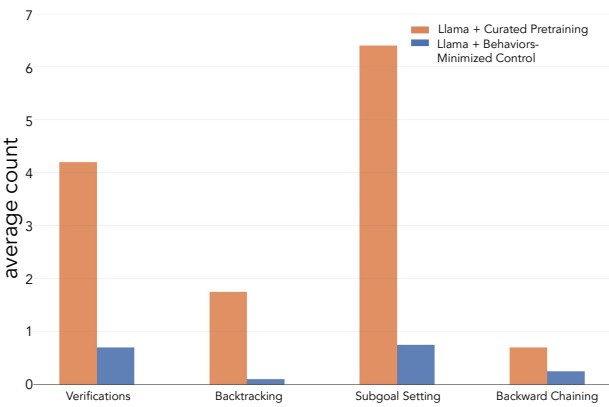

Figure 13: **Transfer of Behaviors to Question Answering.** Average frequency of four cognitive behaviors (verifications, backtracking, subgoal setting, and backward chaining) observed when solving GPQA questions. Comparison between a Llama model trained with curated pretraining data (orange) that amplified these behaviors in mathematical reasoning versus a behavior-minimized control model (blue).

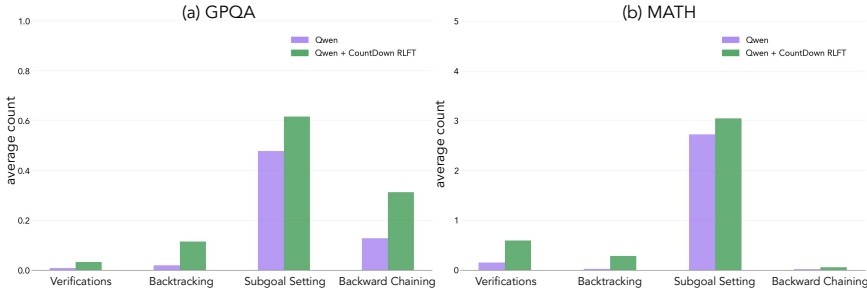

Figure 14: **Transfer of Behaviors to Question Answering learned through RL.** Average frequency of four cognitive behaviors (verifications, backtracking, subgoal setting, and backward chaining) observed when solving (a) GPQA and (b) MATH questions. Comparison between a Qwen model RL finetuned on Countdown (green) that amplified these behaviors in mathematical reasoning versus the base Qwen model (purple).

and MATH each. The results (see Fig. 14) demonstrate a similar accentuation of learned behaviors, with the RL finetuned model demonstrating higher frequencies of all four cognitive behaviors. Surprisingly, we find a performance increase from 38% to 50% with our prompt format and 4-shot 44% to zero-shot 50% with Qwen's 4-shot prompt format on the MATH dataset. Similar to the previous finding, we observe substantial increases in backtracking (0.15 to 0.59) and verification (0.02 to 0.29) in MATH and on GPQA, all four behaviors show meaningful increases: verification (0.008 to 0.03), backtracking (0.019 to 0.114), subgoal setting (0.478 to 0.616), and backward chaining (0.128 to 0.313). Notably, subgoal setting and backward chaining were *not* demonstrated by the Countdown RL finetuned Qwen model on Countdown (see Fig. 1 (Middle)) but despite this, these particular cognitive behaviors show accentuated activations post RL finetuning. These results taken together suggest that cognitive behaviors induced in the context of mathematical reasoning have a larger signature across other question-answering domains and beyond the training domain.

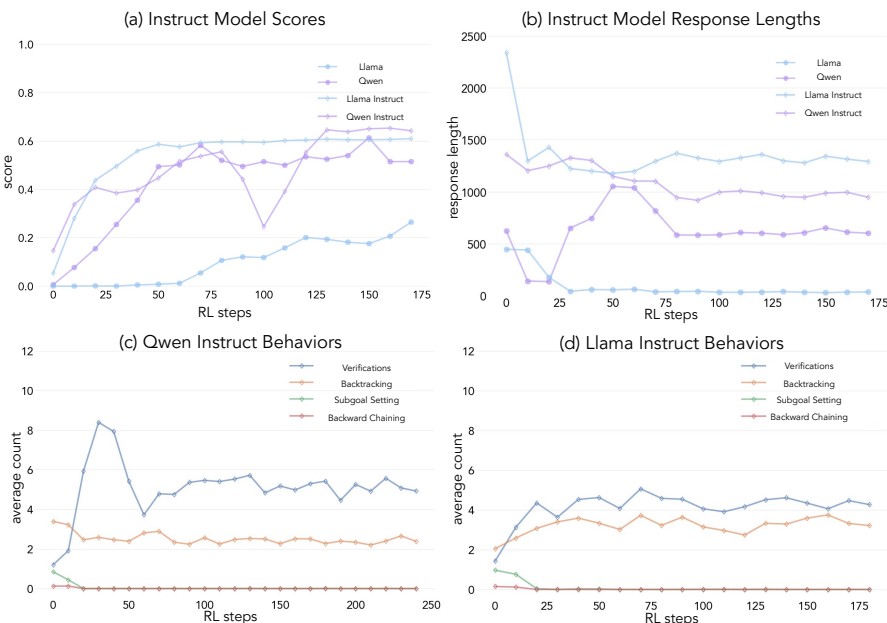

Figure 15: **Analysis of the effect of instruction-tuning on the four cognitive behaviors.** Plots compare (a) score and (b) response lengths between base models and their instruction tuned counterparts; and plotting the four key cognitive behaviors through the course of RL on Countdown with (c) Qwen2.5-3B-Instruct and (d) Llama3.2-3B-Instruct.

## I  Results for Instruction-tuned Models

We investigated whether instruction tuning affects the emergence and amplification of cognitive behaviors during reinforcement learning (RL). Instruction tuning teaches models to reliably follow user instructions, but its impact on cognitive behaviors through RL is unclear. Toward this, we ran our behavioral probes on instruction-tuned variants of Llama-3.2-3B and Qwen-2.5-3B models on Countdown through RL training, following the same methodology applied to their base model counterparts. Our results (see Fig. 15) show that instruction-tuned models from both families nearly saturate task score and use more of their token budgets than the base models. More significantly, we observed the emergence and progressive strengthening of key cognitive behaviors—particularly verification and backtracking. These findings suggest that instruction-tuning datasets not only teach models to follow instructions but also contain examples of the cognitive behaviors that then surface during subsequent RL finetuning.

## J  Behavioral Analysis for Larger Models

We conducted scaling experiments to determine whether our findings generalize across model sizes and to investigate the role of scale in cognitive behavior emergence. Our initial results revealed a striking difference between model families: Qwen models consistently developed sophisticated cognitive behaviors during RL training, while Llama models showed no evidence of such behavioral emergence. This disparity raised the question about whether the observed differences were artifacts of the specific 3B parameter scale or represented fundamental data differences between the model families. To address this question systematically, we extended our analysis to larger model variants, training Qwen2.5-14B and Llama3.2-8B using GRPO on Countdown. The results (see Fig. 16) provided compelling evidence that our earlier findings are robust across model scales. Qwen2.5-14B demonstrated clear and consistent emergence of three of the four cognitive behaviors we identified with patterns similar to those observed in Qwen2.5-3B. Conversely, Llama3.1-8B failed to exhibit any meaningful cognitive behaviors throughout RL training, despite its substantial

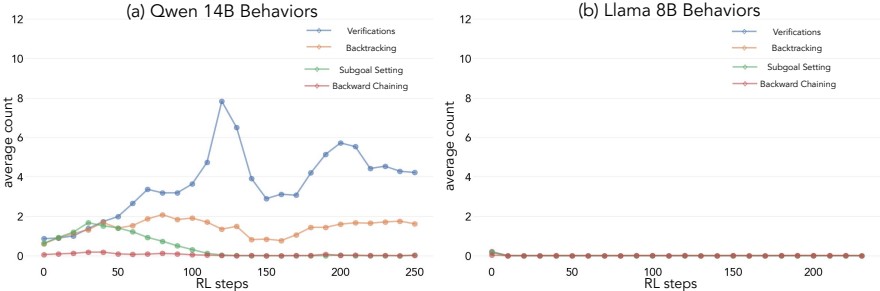

Figure 16: **Analysis of the effect of scaling on the four cognitive behaviors.** Plotting the four key cognitive behaviors through the course of RL training for (a) Qwen2.5-14B and (b) Llama3.1-8B on Countdown.

parameter increase over Llama3.2-3B. The results strongly indicate that model scale alone is insufficient for developing cognitive behaviors during RL training. Instead, the capacity for cognitive behavior emergence appears to be determined by whether the models are exposed to the cognitive behaviors through the course of their pretraining, and that these capacities cannot be reliably induced through scaling alone.

