# OpenReview forum: "Cognitive Behaviors that Enable Self-Improving Reasoners, or, Four Habits of Highly Effective STaRs"
_colmweb.org/COLM/2025/Conference — COLM 2025_

### Official Review · Reviewer_1iHv · 2025-05-06

**Rating:** 7
**Confidence:** 5
**Ethics Flag:** 1

**Summary:**

This paper investigates the behaviors of two models Qwen2.5-3B and Llama3.2-3B  when training with RL on the task of Countdown. It introduces an analysis framework based on four cognitive  behaviors. It is found that Qwen naturally exhibits these behaviors, while LLaMA requires behavior priming via finetuning on specifically designed data. Further experiments show that amplifying LLaMA's behaviors in the math domain can also align its behavior development curve with that of Qwen.

**Questions To Authors:**

1. If Qwen is further trained on the math data that explicitly contains cognitive behaviors, will this amplify its existing behaviors or potentially conflict with its natural behaviors?

2. Appendix H shows that models trained on math data generalize to GPQA in terms of cognitive behaviors, but fail to improve performance. Can the cognitive behaviors developed during RL on the Countdown task transfer to other domains, such as math? If so, do they lead to any performance gains due to these behaviors?

3. In Figure 8(c), there is a noticeable increase in verification behavior between steps 50–90. Could you please elaborate on why the model behaves this way? Have you checked the outputs during these steps to explain the spike?

**Reasons To Accept:**

1. The paper presents a timely and detailed analysis of cognitive behaviors in language models trained via RL on the Countdown task, offering insights into model-specific reasoning patterns.
2. The paper is well written, clearly structured and easy to follow.
3. The proposed framework of four cognitive behaviors provides a useful lens for understanding and analyzing models' self-improvement capabilities.

**Reasons To Reject:**

1. All experiments are conducted on 3B models. It remains unclear whether the observed behaviors generalize to larger or smaller model sizes.
2. The analysis is restricted to base models. Given that instruct models (such as Qwen2.5-3B-Instruct) are trained on more diverse datasets, their cognitive behaviors may differ.

---

> ### Author Response · Authors · 2025-06-03
>
> Thank you for finding our paper "well written, clearly structured and easy to follow" and recognizing our framework's utility.
> > All experiments are conducted on 3B models
>
> We've added experiments with Llama-3.1-8B and Qwen-2.5-14B. Results show consistent patterns — Qwen-14b naturally exhibits target behaviors and improves while Llama-8b struggles and plateaus early in training — confirming our findings scale beyond 3B models.
> ## Model Behaviors Across Training Steps
> ### Step 0
> | Model | Ver | Backtrack | Subgoal | BC |
> |-------|--------------|--------------|---------|-----|
> | Qwen 3B | 0.621 | 0.648 | 0.539 | 0.0654 |
> | Qwen 14B | 0.94 | 0.66 | 0.78 | 0.08 |
> | Llama 3B | 0.1 | 0.195 | 0.29 | 0.03 |
> | Llama 8B | 0.205 | 0.19 | 0.16 | 0.025 |
> ### Step 100
> | Model | Ver| Backtrack | Subgoal | BC |
> |-------|-------|------|----|-----|
> | Qwen 3B | 6.01 | 4.945 | 0.0 | 0.015 |
> | Qwen 14B | 3.37 | 1.6 | 0.305 | 0.05 |
> | Llama 3B | 0.015 | 0.0 | 0.0 | 0.0 |
> | Llama 8B | 0.0 | 0.0 | 0.0 | 0.005 |
> ### Final Step
> | Model | Ver | Backtrack | Subgoal | BC |
> |-------|--------------|--------------|---------|-----|
> | Qwen 3B | 0.27 | 1.74 | 0.0 | 0.02 |
> | Qwen 14B | 4.465 | 1.465 | 0.04 | 0.01 |
> | Llama 3B | 0.0 | 0.0 | 0.0 | 0.0 |
> | Llama 8B | 0.0 | 0.0 | 0.0 | 0.0 |
> > instruct models are trained on more diverse datasets, their cognitive behaviors may differ.
>
> We've added analysis of Qwen-2.5-3B-Instruct and Llama-3.2-3B-Instruct: Instruct models show interesting differences: Qwen-Instruct reaches higher performance (~70%) faster than the base version, while Llama-Instruct explores the behaviors necessary for improvement and outperforms the base model significantly achieving ~62% vs ~37% accuracy.
> ## Behaviors w/ Training Steps (Base vs Instruct)
> ### Step 0
> | Model | Ver | Backtrack | Subgoal | BC |
> |-------|--------------|--------------|---------|-----|
> | Qwen 3B | 0.621 | 0.65 | 0.54 | 0.07 |
> | Qwen 3B-Instruct | 1.21 | 3.39 | 0.85 | 0.12 |
> | Llama 3B | 0.1 | 0.19 | 0.29 | 0.03 |
> | Llama 3B-Instruct | 1.43 | 2.05 | 0.97 | 0.15 |
> ### Step 100
> | Model | Ver | Backtrack | Subgoal | BC |
> |-------|--------------|--------------|---------|-----|
> | Qwen 3B | 6.01 | 4.945 | 0.0 | 0.015 |
> | Qwen 3B-Instruct | 5.465 | 2.57 | 0.0 | 0.0 |
> | Llama 3B | 0.015 | 0.0 | 0.0 | 0.0 |
> | Llama 3B-Instruct | 4.06 | 3.15 | 0.01 | 0.0 |
> ### Final Step
> | Model | Ver | Backtrack | Subgoal | BC |
> |-------|--------------|--------------|---------|-----|
> | Qwen 3B | 0.27 | 1.74 | 0.0 | 0.02 |
> | Qwen 3B-Instruct | 4.925 | 2.385 | 0.0 | 0.0 |
> | Llama 3B | 0.0 | 0.0 | 0.0 | 0.0 |
> | Llama 3B-Instruct | 4.27 | 3.215 | 0.0 | 0.0 |
>
> > Can the cognitive behaviors developed during RL on the Countdown task transfer to other domains, such as math?
>
> RL training on Countdown transfers cognitive behaviors to other reasoning domains. After 100 training steps, Qwen-2.5-3B improved on Countdown (38%→50%) and showed amplified cognitive behaviors on unrelated tasks. On MATH problems: backtracking increased 0.15→0.59, verification 0.02→0.285. On GPQA, all four behaviors increased: verification 0.008→0.033, backtracking 0.019→0.114, subgoal setting 0.478→0.616, backward chaining 0.128→0.313. This suggests Countdown RL instills generalizable reasoning strategies.
> ### MATH
>
> | Model | Accuracy |
> |-------|----------|
> | Qwen2.5-3B (our prompt) | 0.38 |
> | Qwen2.5-3B (qwen’s 4-shot) | 0.42 |
> | Countdown RL Step 50 | 0.44 |
> | Countdown RL Step 100 | 0.49 |
>
> | Behavior | Step 0 | Step 50 | Final Step|
> |----------|--------|----|------|
> | Backtrack | 0.15 | 0.54 | 0.59 |
> | Ver | 0.02 | 0.54 | 0.29 |
> | Subgoals | 2.73 | 2.65 | 3.04 |
> | BC | 0.02 | 0.1 | 0.06 |
>
> ### GPQA
> | Behavior | Base | Countdown Step 100 |
> |------|------|------|
> | Backtrack | 0.01875 | 0.11375 |
> | Ver | 0.0075 | 0.0325 |
> | Subgoals | 0.4775 | 0.61625 |
> | BC | 0.1275 | 0.3125 |
>
> > If Qwen is further trained on the math data … will this amplify its existing behaviors or potentially conflict with its natural behaviors?
>
> We expect that priming the model on more math data will amplify the behaviors similar to countdown where priming the base Qwen model amplifies the existing behaviors.
>
> > there is a noticeable increase in verification behavior between steps 50–90. Could you please elaborate on why the model behaves this way?
>
> The model starts explicitly writing down verification statements in its responses during these steps. Such as:
>
> “55 + 5 - 15 = 45 (too low)
>
> 55 - 5 - 15 = 35 (still too low)
>
> I still did not reach the target.
> I will have to see if I have missed anything..”
>
> Later in training, it starts backtracking without verifying its outputs explicitly in text.
>
> “Let's try different combinations:
>
>  6 + 43 + 13 + 99 = 161
>
> 6 + 43 - 13 + 99 = 135...”
>
> We will add reasoning trajectories over training steps to the appendix to make this clearer.
> We appreciate your feedback and will incorporate it. Please also see the general response (above) and let us know if you have any other questions.

---

> > ### Comment · Reviewer_1iHv · 2025-06-09
> >
> > Thank you for the detailed response! Adding the additional analysis definitely helps strengthen the paper. I’ll raise my score to a 7.

---

### Official Review · Reviewer_ZG7V · 2025-05-10

**Rating:** 7
**Confidence:** 5
**Ethics Flag:** 1

**Summary:**

The paper investigates why some language models (LMs) improve substantially under reinforcement learning (RL), while others plateau. Focusing on two models—Qwen-2.5-3B and Llama-3.2-3B—the authors identify four cognitive behaviors (verification, backtracking, subgoal setting, and backward chaining) as critical for enabling self-improvement. Qwen naturally exhibits these behaviors, leading to superior performance, while Llama does not unless primed. Through controlled experiments, the study demonstrates that even incorrect solutions containing the right reasoning patterns can boost performance, suggesting that exhibiting these behaviors is more important than correctness. Further, the authors show that pretraining on behavior-rich data enables Llama to match Qwen’s RL-driven improvement, establishing a causal link between initial cognitive behaviors and RL efficacy.

**Reasons To Accept:**

- The paper introduces a clear and testable framework for analyzing cognitive behaviors, offering a principled way to study emergent reasoning in LMs beyond task accuracy.

- The use of carefully controlled priming and pretraining experiments, including behavioral classifiers and ablative baselines (e.g., empty CoTs), supports strong causal claims regarding the role of reasoning behaviors.

**Reasons To Reject:**

- The primary testbed (the Countdown game) is narrow and highly structured. Although it's suitable for isolating reasoning behaviors, it remains unclear whether findings generalize to more diverse domains such as math reasoning, code synthesis and other reasoning domains.

- While the authors focus on four cognitive behaviors—verification, backtracking, subgoal setting, and backward chaining—they do not provide a principled rationale (e.g., from cognitive science or computational learning theory) for selecting these particular behaviors. Further, while behaviors are examined individually and in combinations, the work doesn't deeply explore how these behaviors interact or conflict.

- The paper’s comparisons are limited to Qwen-2.5-3B and Llama-3.2-3B.  Without broader coverage: It’s unclear whether the observed gaps are due to behavior differences or opaque model-specific idiosyncrasies. I understand that adding further experiments on larger scale models might be infeasible for an academic project, I would not deem this point a true reason to reject. If the authors could add more experiments on larger scale models (>7B, e.g, 14B or 32B), I'd consider raising my score.

---

> ### Author Response · Authors · 2025-06-03
>
> Thank you for your thoughtful review and recognition that our work provided a “principled way to study emergent reasoning in LMs beyond task accuracy.”
>
> > If the authors could add more experiments on larger scale models (>7B, e.g, 14B or 32B), I'd consider raising my score
>
> We've added experiments with Llama-3.1-8B and Qwen-2.5-14B. Results show consistent patterns — Qwen-14b naturally exhibits target behaviors and improves while Llama-8b struggles and plateaus early in training — confirming our findings scale beyond 3B models.
> ## Model Behaviors Across Training Steps
> ### Step 0
> | Model | Verification | Backtracking | Subgoal | BC |
> |-------|--------------|--------------|---------|-----|
> | Qwen 3B | 0.621 | 0.648 | 0.539 | 0.0654 |
> | Qwen 14B | 0.94 | 0.66 | 0.78 | 0.08 |
> | Llama 3B | 0.1 | 0.195 | 0.29 | 0.03 |
> | Llama 8B | 0.205 | 0.19 | 0.16 | 0.025 |
> ### Step 100
> | Model | Verification | Backtracking | Subgoal | BC |
> |-------|--------------|--------------|---------|-----|
> | Qwen 3B | 6.01 | 4.945 | 0.0 | 0.015 |
> | Qwen 14B | 3.37 | 1.6 | 0.305 | 0.05 |
> | Llama 3B | 0.015 | 0.0 | 0.0 | 0.0 |
> | Llama 8B | 0.0 | 0.0 | 0.0 | 0.005 |
> ### Final Step
> | Model | Verification | Backtracking | Subgoal | BC |
> |-------|--------------|--------------|---------|-----|
> | Qwen 3B | 0.27 | 1.74 | 0.0 | 0.02 |
> | Qwen 14B | 4.465 | 1.465 | 0.04 | 0.01 |
> | Llama 3B | 0.0 | 0.0 | 0.0 | 0.0 |
> | Llama 8B | 0.0 | 0.0 | 0.0 | 0.0 |
>
> We've also added analysis of Qwen-2.5-3B-Instruct and Llama-3.2-3B-Instruct: Instruct models show interesting differences: Qwen-Instruct reaches higher performance (~70%) faster than the base version, while Llama-Instruct explores the behaviors necessary for improvement and outperforms the base model significantly achieving ~62% vs ~37% accuracy. This suggests instruction tuning partially compensates for missing base behaviors.
>
> ## Model Behaviors Across Training Steps (Base vs Instruct)
> ### Step 0
> | Model | Verification | Backtracking | Subgoal | BC |
> |-------|--------------|--------------|---------|-----|
> | Qwen 3B | 0.621 | 0.648 | 0.539 | 0.0654 |
> | Qwen 3B-Instruct | 1.205 | 3.39 | 0.845 | 0.12 |
> | Llama 3B | 0.1 | 0.195 | 0.29 | 0.03 |
> | Llama 3B-Instruct | 1.425 | 2.05 | 0.97 | 0.15 |
> ### Step 100
> | Model | Verification | Backtracking | Subgoal | BC |
> |-------|--------------|--------------|---------|-----|
> | Qwen 3B | 6.01 | 4.945 | 0.0 | 0.015 |
> | Qwen 3B-Instruct | 5.465 | 2.57 | 0.0 | 0.0 |
> | Llama 3B | 0.015 | 0.0 | 0.0 | 0.0 |
> | Llama 3B-Instruct | 4.06 | 3.15 | 0.01 | 0.0 |
> ### Final Step
> | Model | Verification | Backtracking | Subgoal | BC |
> |-------|--------------|--------------|---------|-----|
> | Qwen 3B | 0.27 | 1.74 | 0.0 | 0.02 |
> | Qwen 3B-Instruct | 4.925 | 2.385 | 0.0 | 0.0 |
> | Llama 3B | 0.0 | 0.0 | 0.0 | 0.0 |
> | Llama 3B-Instruct | 4.27 | 3.215 | 0.0 | 0.0 |
>
> > they do not provide a principled rationale (e.g., from cognitive science or computational learning theory
>
> We address this concern in the general response above.
>
> > The primary testbed (the Countdown game) is narrow and highly structured.
>
> We address this concern in the general response above.
>
> We appreciate your feedback and will incorporate it. Please also see the general response (above) and let us know if you have any other questions.

---

> > ### Comment · Reviewer_ZG7V · 2025-06-03
> >
> > Thanks for adding the experiments for larger-scale models. I'll raise my score and lean towards the acceptance of the paper.

---

### Official Review · Reviewer_CcUW · 2025-05-12

**Rating:** 10
**Confidence:** 5
**Ethics Flag:** 1

**Summary:**

This paper investigates the fundamental mechanisms that enable language models to improve their reasoning capabilities, specifically learn self-improving, through reinforcement learning (RL). The authors focus on two 3B-parameter models, Qwen-2.5-3B and Llama-3.2-3B, for comparison when trained on the Countdown game.
The research identifies four key cognitive behaviors critical to effective problem-solving: verification (error-checking), backtracking (abandoning failing approaches), subgoal setting (breaking down complex problems), and backward chaining (reasoning from desired outcomes to initial inputs). Surprisingly, the study finds that Qwen naturally exhibits these behaviors, while Llama does not, leading to significant performance disparities during RL training.
Crucially, when Llama was primed with datasets emphasizing these cognitive behaviors or pretrained on OpenWebMath dataset that amplifies such behaviors, its performance approached that of Qwen and presents desired test-time scaling behaviors. This suggests that a model's initial reasoning capabilities fundamentally determine its potential for self-improvement.

**Reasons To Accept:**

- This paper demystifies the long CoT training, performing a rigorous ablation study to present a reproducible recipe on how to train a o1-like model: priming LLMs with cognitive patterns as priors for RL.
- The experimental setup is rigorous and the results are convincing, even though tasks and models are a bit limited.

**Reasons To Reject:**

- How the authors identified four cognitive behaviors looks a bit arbitrary.
- They only focus on Countdown, a task that may be easier to elicit self-reflection as there is an explicit verification signal (if the calculated result equals target), which is somehow limited.

---

> ### Author Response · Authors · 2025-06-03
> **We address concerns in the general response above**
>
> Thank you for your positive reading and recognition that our work "demystifies the long CoT training" and presents "a reproducible recipe on how to train a o1-like model."
>
> > How the authors identified four cognitive behaviors looks a bit arbitrary.
>
> Please see the general response above where we address this concern.
>
> We address this in the general response above.
>
> > only focus on Countdown … here is an explicit verification signal (if the calculated result equals target), which is somehow limited.
>
> We address this in the general response above.
>
> We appreciate your feedback and will incorporate it. Please also see the general response (above) and let us know if you have any other questions.

---

> > ### Comment · Reviewer_CcUW · 2025-06-03
> > **Thanks for your response**
> >
> > Thanks for your response. This paper is one of the most important works in the post-R1 fever and gives a very fundamental undertsanding on this phenomenon, which profoundly impacts the research community. As my major concerns are addressed, I will raise my score to 10 because in my view this is definitely one of the best papers in COLM.

---

### Official Review · Reviewer_UQGC · 2025-05-18

**Rating:** 6
**Confidence:** 2
**Ethics Flag:** 1

**Summary:**

This paper dives into a curious difference observed when training language models to improve using reinforcement learning: some models get significantly better, while others just sort of hit a wall. The authors noticed this particularly when comparing Qwen-2.5-3B and Llama-3.2-3B on a math game called Countdown, where Qwen improved dramatically and Llama didn't, even with the same training. They wondered what intrinsic qualities in a model allow for this kind of self-improvement.

**Questions To Authors:**

How would this be like if you run it on 70B+ or MoE model?

**Reasons To Accept:**

I think one of the paper's main strengths is identifying and systematically studying these specific cognitive behaviors. It moves beyond just looking at final performance and provides a structured framework for analyzing how models attempt to solve problems. The experimental design, particularly the priming experiments and the finding that incorrect solutions with proper reasoning patterns still enable improvement, is quite insightful and provides strong evidence for their hypothesis about the importance of behaviors over just correctness. The connection back to pretraining data is also a valuable contribution, suggesting ways to engineer models with better self-improvement potential from the ground up.

**Reasons To Reject:**

N/A

---

> ### Author Response · Authors · 2025-06-03
>
> Thank you for your thoughtful review of our work. We are glad that you found our framework and systematic study of behaviors to be insightful.
>
> > How would this be like if you run it on 70B+ or MoE model?
>
> While 70B+ models require computational resources beyond our current access, we have added experiments with Llama-3.1-8B and Qwen-2.5-14B models. We observe similar patterns: Qwen-14B shows natural emergence of verification and backtracking behaviors, while Llama-8B, similar to 3B plateaus early in training. The behavioral differences persist at larger scales, suggesting our findings are not artifacts of model size but dependent on the initial behaviors explored by the model.
> ## Model Behaviors Across Training Steps
> ### Step 0
> | Model | Verification | Backtracking | Subgoal | BC |
> |-------|--------------|--------------|---------|-----|
> | Qwen 3B | 0.62 | 0.65 | 0.54 | 0.065 |
> | Qwen 14B | 0.94 | 0.66 | 0.78 | 0.08 |
> | Llama 3B | 0.1 | 0.20 | 0.29 | 0.03 |
> | Llama 8B | 0.20 | 0.19 | 0.16 | 0.03 |
> ### Step 100
> | Model | Verification | Backtracking | Subgoal | BC |
> |-------|--------|--------|-----|-----|
> | Qwen 3B | 6.0 | 4.95 | 0.0 | 0.02 |
> | Qwen 14B | 3.37 | 1.6 | 0.305 | 0.05 |
> | Llama 3B | 0.02 | 0.0 | 0.0 | 0.0 |
> | Llama 8B | 0.0 | 0.0 | 0.0 | 0.01 |
> ### Final Step
> | Model | Verification | Backtracking | Subgoal | BC |
> |-------|--------|-------|-----|-----|
> | Qwen 3B | 0.27 | 1.74 | 0.0 | 0.02 |
> | Qwen 14B | 4.465 | 1.465 | 0.04 | 0.01 |
> | Llama 3B | 0.0 | 0.0 | 0.0 | 0.0 |
> | Llama 8B | 0.0 | 0.0 | 0.0 | 0.0 |
>
> We will include the complete results and analyses in the appendix.
> We appreciate your feedback and will incorporate it. Please also see the general response (above) and let us know if you have any other questions.

---

> ### Author Response · Authors · 2025-06-10
>
> As the discussion period draws to a close, please let us know if there are any remaining questions or concerns you'd like us to address. Also, let us know whether we successfully answered your initial questions about models of different scales.

---

### Author Response · Authors · 2025-06-03
**General Response**

We thank all reviewers for their thoughtful and constructive feedback. We are pleased that reviewers found our work "timely" (`1iHv`), providing "a rigorous ablation study to present a reproducible recipe on how to train a o1-like model" (`CcUW`), with "strong causal claims" (`ZG7V`) and “insightful” (`UQGC`). We're grateful for the reviewers' recognition that our paper is "well written, clearly structured and easy to follow" (`1iHv`).

In response to the reviewers' valuable suggestions, we will add:

## New experiments and analyses:

- **Larger model experiments** (`UQGC`, `ZG7V`, `1iHv`): We added results with Llama-3.1-8B and Qwen-2.5-14B models.  Results show consistent patterns — Qwen-14b naturally exhibits target behaviors and improves while Llama-8b struggles and plateaus early in training — confirming our findings scale beyond 3B models.

- **Instruct model analysis** (`UQGC`, `ZG7V`, `1iHv`): Tested Qwen-2.5-3B-Instruct and Llama-3.2-3B-Instruct to examine how instruction tuning affects cognitive behaviors. Instruct models are better at exploring these behaviors.

- **Transfer of RL training on Countdown** (`1iHv`): We demonstrate that cognitive behaviors developed during RL on Countdown transfer to other reasoning domains. After 100 steps of RL training on Countdown, Qwen-2.5-3B shows not only improved task performance (38%→50% with our prompt format, 4-shot 44%→0 shot-50% with Qwen's 4-shot format) but also exhibits amplified cognitive behaviors when evaluated on unrelated tasks. On MATH problems, we observe substantial increases in backtracking (0.15→0.59) and verification (0.02→0.29). Similarly, on GPQA, all four behaviors show meaningful increases: verification (0.008→0.03), backtracking (0.019 →0.114), subgoal setting (0.478→0.616), and backward chaining (0.128→0.313). These results suggest that RL training on Countdown instills reasoning strategies that transfer beyond the training domain.

## Discussion on our choice of behaviors (addressing `CcUW`, `ZG7V`):

We acknowledge this concern and will expand Section 3.2 to provide principled rationale for our behavior selection:

- **Theoretical grounding**: These behaviors are well-established in cognitive science literature on expert problem-solving (Newell & Simon, 1971; Polya, 1945). Further, all our behaviors are drawn from classical planning Fikes, R. E., & Nilsson, N. J. (1971).
- **Computational necessity**: Each behavior addresses a fundamental limitation of linear reasoning: verification (error detection), backtracking (error correction, exploration), subgoal setting (problem decomposition), and backward chaining (goal-directed planning)
- **Domain generality**: Unlike domain-specific strategies, these behaviors apply across reasoning tasks
- **Ease of identification**: These behaviors can be reliably identified in text outputs.

We will update our draft to make this clearer.

## Benefits and Limitations of Countdown (`CcUW`, `ZG7V`):

We chose Countdown as our primary testbed for a few reasons:

First, it provides a controlled environment to test our key hypothesis: what behaviors are necessary for self-improvement with RL? The game presents a challenging search problem with a large branching factor and an inherent look-ahead problem, capturing the complexity of planning tasks while remaining tractable.

Importantly, Countdown offers advantages over seemingly richer domains like MATH or code generation. Current models are often over-optimized for these tasks during pretraining, making it difficult to study post-training learning dynamics. In contrast, models have minimal exposure to Countdown, providing a cleaner experimental setting to observe emergent behaviors during RL.
While we acknowledge that verification in Countdown (as pointed out by `CcUW`) is more straightforward than in open-ended domains, this simplicity is actually advantageous for our core investigation. It allows us to isolate the cognitive behaviors that enable self-improvement i.e., the "aha moment" in learning, without confounding factors from domain knowledge or ambiguous success criteria. The straightforward forward inference steps in Countdown (simple arithmetic) let us clearly observe when and how models develop more sophisticated search strategies.

We will expand the discussion to recognize this focused approach as a limitation and discuss paths toward broader domains in our revised discussion section. However, we believe establishing these fundamental principles in a controlled setting provides essential groundwork for understanding self-improvement in more complex reasoning tasks.

We hope that these revisions help to address the key points raised by the reviewers and strengthen the presentation of our work. Thank you again for the constructive feedback that has helped us to improve the paper.

---

> ### Author Response · Authors · 2025-06-03
> **References**
>
> Fikes, R. E., & Nilsson, N. J. (1971). STRIPS: A new approach to the application of theorem proving to problem solving. Artificial intelligence, 2(3-4), 189-208.
>
> Simon, H. A., & Newell, A. (1971). Human problem solving: The state of the theory in 1970. American psychologist, 26(2), 145.
>
> Polya, G. (1945). How to solve it: A new aspect of mathematical method. In How to solve it. Princeton university press.

---

### Decision · Program_Chairs · 2025-07-08

**Decision:**

Accept

**Comment:**

The paper identifies that certain models are inherently more amenable to improved reasoning via RL finetuning due to the pretraining data or priming. The reviewers generally agree that the work is principled and systematic, providing a rigorous framework and axes on which to evaluate these reasoning capabilities that leads to the clear and convincing conclusions. The authors additionally addressed the concerns relating to two potential limitations (model sizes, categorization of reasoning), with only one (choice of task) which remains somewhat unaddressed. Nonetheless, the reviewers believe this study is already sufficiently well-structured and informative that I recommend acceptance.